# Discrepancies between Expected and Actual Implementation: The Process Evaluation of PERS Integration in Nursing Homes

**DOI:** 10.3390/ijerph17124245

**Published:** 2020-06-14

**Authors:** Fangyuan Chang, Andrea Eriksson, Britt Östlund

**Affiliations:** Department of Biomedical Engineering and Health Systems, School of Engineering Sciences in Chemistry, Biotechnology and Health, KTH Royal Institute of Technology, SE-142 58 Stockholm, Sweden; andrea4@kth.se (A.E.); brittost@kth.se (B.Ö.)

**Keywords:** health technology services, process evaluation, care practices, implementation, Personal emergency response system, nursing homes

## Abstract

Recent studies prove that when implementing new technology technology-driven and one-size-fits-all approaches are problematic. This study focuses on the process of implementing personal emergency response system (PERS) at nursing homes. The aim is to understand why the implementation of PERS has not met initial expectations. Multiple methods were used in two Swedish nursing homes, including document analysis, questionnaires (*n* = 42), participant observation (67 h), and individual interviews (*n* = 12). A logic model was used to ascertain the discrepancies that emerged between expected and actual implementation, and the domestication theory was used to discuss the underlying meanings of the discrepancies. The discrepancies primarily focused on staff competence, system readiness, work routines, and implementation duration. Corresponding reasons were largely relevant to management issues regarding training, the procurement systems, individual and collective responsibilities as well as invisible work. The uptake of technology in daily practice is far more nuanced than a technology implementation plan might imply. We point out the importance of preparing for implementation, adjusting to new practices, and leaving space and time for facilitating implementation. The findings will be of use to implementers, service providers, and organizational managers to evaluate various measures in the implementation process, enabling them to perform technology implementation faster and more efficiently.

## 1. Introduction

There is evidence that various difficulties exist in technology implementation at nursing homes [1,2]. Interventionist approaches to testing new technologies have been criticized for hampering practical impact through a disproportionate focus on technology as the primary agent of change, and not taking into consideration the context of care practice [3,4,5,6,7]. Many projects fail in the pilot stage. And those that succeed to embed technologies in routine use are usually on a smaller scale, or take longer than expected to implement [8,9,10,11]. This suggests that, despite the growing rhetoric supporting the role of technology in care, there remain significant challenges to its uptake in daily practice.

The personal emergency response system (PERS) is one of the most basic technologies implemented in nursing homes and for elderly care. It was developed in the 1970s and has been recognized as a fundamental primary care tool in western countries [12]. It usually consists of a telecare system and alarm buttons in the form of pendants or wristbands. An important function of the system is to provide remote and in-time communication between residents and their assistant nurses. By clicking the alarm button, residents can ask for help from their assistant nurses without spatial or temporal limitations. Hence, the system has the potential to ensure the security and safety of residents, and provide well-organized working environments for assistant nurses [13]. With such promising benefits, many western countries incorporate PERS in policy as a means of social support and promote its implementation. As a leading country in technological innovation, Sweden has an incredible PERS penetration rate, with over 200,000 users in 2015 [14].

However, the uptake of PERS into daily practice has been uneven and slow, despite evidence that it can be used effectively to monitor and support residents, and appears to have provided promising solutions for eldercare services for decades [15,16]. The Swedish government’s goal was that all old PERS should be replaced at the end of 2016, as most were installed decades ago and have many technical problems, such as bad connection and poor compatibility [14]. Nevertheless, although extensive efforts have been made over the past few years, the date for achieving this goal was postponed indefinitely due to poor implementation yields. Similar situations were also reported across other countries [17,18].

Previous studies in technology implementation focus predominantly on the impacts of certain technologies after implementation, and the corresponding causes [13,19,20,21,22]. These studies tell us a good deal about the effectiveness of specific technology in a relatively specific context, but much less about how to translate the findings and how they might be applied elsewhere. As illustrated by recent studies, more investigations are required to explain why the same technology cannot achieve similar implementation outcomes [23]. Consequently, scholars appeal for a shift of point to the implementation process through which outcomes occur. There are indeed several studies concerning the implementation process of PERS. These studies contribute valuable discussion on actors involved [24,25,26], use contexts [27,28] or technology design [9,29]. For example, Randi [24] examined different actors involved in the implementation process, and found their expectations on the PERS in use are multidimensional. Hanley and colleagues [29] noted system usability challenges healthcare professionals in the implementation process despite the fact that both healthcare professionals and their patients have positive technology attitudes. Andreassen and colleagues [30] highlighted the adaptation of technology in routine care practices by exploring why some PERS implementation projects survive even if there is an apparent lack of success. However, little is known about the role of the implementation plans and interventions in terms of the uptake of PERS in daily practices.

To fill this gap, this study aims to understand why the implementation of PERS has not met initial expectations. Research questions include:What are the discrepancies between expected and actual PERS implementation (RQ1)?How and why do the discrepancies between expected and actual PERS implementation occur (RQ2)?

The findings will be of use to implementers, service providers, and organizational managers to evaluate various measures in the implementation process, enabling them to perform technology implementation faster and more efficiently.

## 2. Theoretical Framework

Various theories or models demonstrate how implementation plans facilitate change. One of the most popular is the logic model [31]. It usually includes different components of an implementation project (i.e., inputs, activities, mediated outcomes, and long-term outcomes). Using a simplified graphic description, a logic model can present the relationship between different implementation measures [32], and clarify the underlying assumptions about how these implementation measures are expected to work [33]. In this way, the logic model provides a visual summary of implementation project planning, conducting, and evaluation. Generally, the purpose of logic models is to track implementation progress and evaluate implementation measures. Besides providing a guide for tracking how an implementation project and its components evolve over time, a logic model can also help specify important implementation components that need to be measured during the evaluation, thus answering crucial questions that relate to the priorities of resource allocation [34]. Hence, the logic model provides a simplified visual representation illustrating different components of an implementation project and the relationships between them. This study utilized the logic model to clarify the initial project expectations and provide a clear structure for monitoring the discrepancies between the expected and actual implementation situation.

Despite these advantages, the logic model has been criticized for its underlying rationalist discourse. In this discourse, implementation is viewed as a linear and standardized process, through which one thing leads to another in predictable and measurable ways [35,36]. The quantifying and formal considerations oversimplify the complexity of reality and make fixed culture and morals as mechanical components that resemble parts of a machinery system. As a result, logic models may fail to account for the complex connections between the implementation project and target contexts, leading to incorrect interpretations [36]. Opposed to the rationalist discourse, scholars in science and technology studies (STS) have called for a rethinking of technology in use from a socio-technical perspective [27,37,38]. So instead of viewing technology as an artefact that can ‘drop into’ routine practices by simply adding expectations, people need to realize technology is an integrated component in a system of care practices.

Silverstone and Hirsch [39] propose domestication theory to understand how technology eventually integrates into care practice during the implementation. The term ‘domestication’ refers to the socialization process of technology that resembles that whereby wild animals become ‘house-trained’ [40]. This process starts when the new technology is employed in daily work. During the process, users transform the technology to make them adaptive and meaningful to the existing particular practices and routines [40]. At the same time, the technology unleashes its inscribed assumptions regarding how it is supposed to work [41,42], and affects its users and shapes the existing practices [38]. As a consequence, how the technology works may differ from its inscribed assumptions, and the performance of a technology may differ in different contexts where it is implemented [43,44,45]. The uptake of new technology in daily practices is thus acknowledged as a complex and unpredictable process through which ‘a technology can shape and mediate social environments and relationships, which can, in turn, shape the functionalities and usage of the technology’ [12]. Hence, technology implementation is an iterative rather than a linear process; technology, users, and existing practices intertwine to co-create a new routine of everyday practices through predictable and unpredictable values, resistance, tension, pride, and refusal. In this study, the domestication theory was used to understand the underlying reasons for different discrepancies, explaining how such discrepancies occur.

## 3. Methods

A case study of the implementation process of PERS was conducted [46]. The logic model was used as an analytic technique to clarify implementation and plans as well as to compare with actual intermediate outcomes and long-term outcomes (ibid., p.127). Data were collected by a variety of methods to develop a comprehensive understanding of the implementation process and to enable a more complete picture of practice [47]. This involved document analysis, in-depth interviews, questionnaire survey as well as observation. Specifically, the document analysis and questionnaire survey were used to widely understand the situation of implementation while interviews and observation were used to deeply understand the discrepancies occurring during the process and participants’ perceptions of the implementation.

### 3.1. Settings

This study was conducted in two Swedish nursing homes. A selection was made from nursing homes in Sweden, to include those that are implementing or intend to implement PERS within half a year. This was conducted by contacting development managers in different municipalities throughout the country. Although most managers showed interest, few nursing homes were eligible. We met leaders of the nursing homes that meet the inclusion criteria, to obtain permission to do the study. Two nursing homes were identified: one in Stockholm and one in Gothenburg. The nursing homes in this study had similarities in terms of size, unit functions, and organizational structure (Table 1).

### 3.2. Data Collection

The data collection lasted six months (August 2019 to January 2020) from pre-implementation to post-implementation. Document analyses and interviews were conducted beforehand to understand implementation expectations, while questionnaires were distributed before and after the implementation to ascertain the actual situation of implementation. Moreover, interviews and observations were performed during the implementation to understand reasons and meanings underpinning identified discrepancies.

As shown in Table 2, participants in this study included nursing home leaders, middle managers, and assistant nurses. In Sweden, assistant nurses may have an exam from college, but the job title is not licensed. They have a formal training and a diploma in terms of providing care at home, in nursing homes or in hospitals [48]. Each individual gave formal consent. After selecting the nursing homes, we started the recruitment process. To recruit leaders and middle managers, we asked their interests when having a face-to-face meeting; to recruit assistant nurses, after receiving all assistant nurses’ consent, we distributed questionnaires where assistant nurses could present their interests in participating in interviews and observations. We first contacted the ones who show their interests, and then used snowball sampling to find other participating assistant nurses. Interviews ended when data saturation was reached [49]. As a result, we included all assistant nurses in the survey, observed middle managers and assistant nurses for 67 h, and interviewed one nursing home leader, four middle managers and seven assistant nurses.

#### 3.2.1. Data Concerning Implementation Expectations (Document Analysis and Interview with Leader)

Reviewed documents included information from the service provider websites, product brochures, instruction manuals, and local plans for PERS implementation.

We used a semi-structured topic guide for the interview with the nursing home leader. Because of time conflicts, only one nursing home leader was interviewed. To ensure our identified implementation expectations and plans fit two nursing homes, the leaders from both nursing homes were asked to confirm the logic model. The interview started with an open question to get a holistic view of the PERS implementation project (i.e., ‘think of the system that you are going to implement, what are your expectations?’). We then asked the leader to reflect on planned implementation strategies (i.e., ‘what are the inputs and activities that aim to achieve the expectations?’), and corresponding outcomes (i.e., ‘what is [the planned activity] used for?’). The interview lasted around 40 min. During the interview, the first author regularly summarized the leader’s responses to ensure the credibility of collected data. We did not tape-record the interview because the leader felt uncomfortable with recording. But the interview notes were taken and were confirmed by the leader.

#### 3.2.2. Data Concerning Actual Implementation (Survey)

In this part of the study, the pre-implementation survey was to understand assistant nurses’ perspectives on the older system which was used in the nursing homes before the new PERS, while the post-implementation survey was to understand their perspectives on the new PERS. The survey response rate was 72% (*n* = 57/79) before the implementation and 60% (*n* = 46/77) after it. In total, we had longitudinal data for 21 individuals (out of 51 assistant nurses who completed the pre-implementation survey and were still working in the nursing homes when we conducted the post-implementation survey). The survey collected information relevant to personal attitudes towards the technology per se and towards the implementation strategies. We chose to focus on these aspects because users with positive perceptions in these aspects are more likely to accept and use the technology efficiently [15,19]. In terms of detailed items, we combined two validated questionnaires to ensure the high reliability of this scale; questions related to personal attitudes towards PERS were from the System Usability Scale [50], and those related to implementation strategies were from the S-NoMAD instrument [51]. This questionnaire included 5-point Likert scale questions (ranging from ‘totally disagree’ as 1 to ‘totally agree’ as 5), closed questions (with options such as ‘yes’ or ‘no’), and open questions (Appendix A). It was pre-tested with three assistant nurses to ensure validity and clearness.

#### 3.2.3. Data Concerning the Occurrence of Discrepancies (Observations and Interviews with Middle Managers and Assistant Nurses)

Before the investigation, the first author spent two weeks becoming familiar with the staff and work structures in the selected nursing homes. This involved accompanying middle managers and assistant nurses working in different units in order to: (1) understand the social and cultural contexts in the nursing homes to decrease the potential bias when collecting and interpreting data, and (2) to build trust with participants, so making the first author ‘invisible’ in the working environment when investigating participants’ real behaviours/ideas.

The observations were conducted twice a week, with each lasting around two to four hours [52]. During the observation, the first author collected extensive field notes about inhibiting or facilitating activities during the PERS implementation process. These included, for example, middle managers’ behaviours around promoting the PERS implementation process, behaviours among assistant nurses in integrating PERS into daily work, interactions between managers and assistant nurses in terms of system problems, and individual responsibilities and roles in PERS implementation. After each observation, the first author reviewed the field notes and added reflective notes.

The interviews with middle managers and assistant nurses lasted 21 to 45 min and were conducted in a separate room. We began with an open question such as ‘I would love to hear your experience with the implementation’. Follow-up questions focused on reflections on the activities aiming to integrate PERS into routine use, perceived responsibilities and received support in this process, and potential solutions for promoting the implementation. During the interview, the first author regularly summarized participants’ responses to ensure the credibility of the collected data. Each participant gave informed consent verbally, and most of them signed on a participant information sheet. With the verbal consent from each participant, we recorded all these interviews and transcribed them verbatim.

### 3.3. Data Analysis

The data analysis consisted of three steps. First, we drew a logic model of implementation strategies and expected outcomes. We then compared this with our survey data to ascertain whether there were discrepancies between expectations and actual implementation, and what they were. Lastly, we used the observation and interview data to explain how the discrepancies occurred. All participants were given a participant identifier.

#### 3.3.1. Logic Model Establishment

From the data collected from implementation documents and the leader’s interview, we extracted text segments and categorized them into inputs, activities, intermediate outcomes, and long-term outcomes. In each category, we merged similar text segments and performed double coding to summarize them into consistent phrases. The qualitative data in ‘Expected intermediate outcomes’ was further categorized into staff competence, system, and work routines according to the content of double coded phrases. The expected long-term outcomes were summarized from the technology instruction manual and were consistent with the leader’s interview. Planned inputs, activities, and intermediate outcomes were summarized from the leader’s interview and local documents, such as training guidelines (Table 3).

In this model, ‘Expected intermediate outcomes’ referred to expected direct outcomes of the inputs and activities; ‘Expected long term outcomes’ referred to expecting long term benefits or disadvantages resulting from intermediate outcomes [53]. According to our research aims, the data analysis primarily focused on the discrepancies that occurred during the implementation process (i.e., the ‘expected intermediated outcomes’ in Table 3).

#### 3.3.2. Discrepancies Identification

Data collected from questionnaires focused on assistant nurses’ actual attitudes towards the new PERS and implementation strategies. Through comparing it with the built logic model, we focused on whether initial project expectations were achieved. For example, one expected intermediate outcome was that all assistant nurses gain adequate knowledge and skills. According to the survey data, over one-fifth of assistant nurses reported their needs to learn more knowledge. We thus confirmed there existed discrepancies regarding getting all staff adequately trained. In terms of the questionnaire data analysis, for Likert scale questions, we averaged the collected ratings; for closed questions, we calculated the percentage of each option; and for open questions, we extracted and summarized key contents of collected answers (Appendix A).

#### 3.3.3. Underlying Reasons for Identified Discrepancies

Based on the logic model, the quantitative data about unanticipated or unsatisfactory implementation consequences was first categorized into staff competence, system, work routine and others; and second, combined with the observational data and interview transcripts. The qualitative data was used to understand the unanticipated or unsatisfactory implementation consequences found in the quantitative data [54]. The qualitative data was reviewed separately, line by line in NVivo 12. Text segments regarding the reasons/occurrences of discrepancies were extracted from relevant lines. We then created codes to summarize the core meanings of the extracted text segments. To ascertain whether a code was appropriately assigned, we compare text segments to segments that had been previously assigned the same code to see whether they reflected the same meaning. Through this process, the themes that different codes belong to evolved inductively. Codes and themes were reviewed by all authors to ensure consistency in coding approach. These themes were then related to the identified discrepancies to understand complex phenomena regarding the occurrence of identified discrepancies and to generate connections and constructs between phenomena [55].

### 3.4. Ethical Considerations

No personal or sensitive information was collected. No ethical permission was thus needed according to Swedish law [56]. In terms of data collection, all participants received both oral and written information about the study. For the survey, each participant was required to sign after reading and hearing the consent information. For the interviews and observations, each participant confirmed informed consent verbally. In terms of data analysis, all participants were given a participant identifier, and transcripts were anonymized to ensure personal privacy.

## 4. Results

As stated in the beginning, the aim of this paper is to explore and explain some of the unanticipated or unsatisfactory consequences of PERS implementation processes. However, it should be noted that though the PERS implementation was not as satisfactory as expected, assistant nurses perceived high usability of the new PERS. Most assistant nurses thought that patient safety and user satisfaction were improved. The positive reflections can be especially observed among staff who took care of residents with severe health problems, as they thought the new PERS provided consistent information that their residents were safe in the nursing home. In addition, most assistant nurses thought many of their residents learned about the use of new PERS quickly.

Table 4 outlines the discrepancies between expected and actual implementation, and corresponding reasons. Briefly, the discrepancies focused on four main domains: limited staff competence, limited system readiness, limited routine fluency, and longer duration of implementation. Corresponding reasons focused mainly on the management issues in relation to staff training, the procurement system, individual and collective responsibilities as well as invisible work.

### 4.1. Staff Competence

#### 4.1.1. Limited Staff Competence

Assistant nurses were not proficient in using the new PERS when performing care, despite the expectation that all assistant nurses could gain adequate knowledge and skills from the planned training session and training materials (Table 4). In both nursing homes, according to our survey, some assistant nurses still thought they needed more knowledge in PERS use, and the training materials were not mentioned as influential. The findings of limited staff competence were consistent with our observation notes. For example, we observed that some assistant nurses kept alarm phones unintentionally muted and missed several alarms because they had no idea that they could adjust the volume to ensure the alarms could be heard.

#### 4.1.2. Reasons for Limited Staff Competence

The reasons why assistant nurses were not proficient in using the new PERS were similar in the two nursing homes. This included complicated training materials and a lack of training on how to apply the system in local contexts (Table 4).

The training materials were not mentioned as influential as expected. First, assistant nurses described the difficulties finding the training materials, as they were “*stored randomly everywhere [01A04]*”. Additionally, there was a discussion on the need for training materials to be digestible and attractive. The difficulties in reading training documents were highlighted, “*They have many official terms that we don’t use daily, and they are not friendly to the audience…not very easy to read [02A02]*”. One middle manager also emphasized the training documents were tediously long, “*Staff want something quick and brief [01B01]*”. Assistant nurses tended to learn from colleagues rather than from distributed training materials. As we observed, they relied on what they learned from their colleagues, as it was quicker and easier. In this case, training materials were mentioned as influential only when these assistant nurses realized something went wrong. For example, some assistant nurses from the same ward had the same incorrect way of using the new PERS. After being reminded of the correct approaches to using the new PERS, they turned to read the training materials. These assistant nurses did acknowledge that merely learning from colleagues might constrain their knowledge in using the new PERS “*we know the shortcomings, but we don’t need to master the system, I mean every detail of the system [02A04]*”. Therefore, they would consult the training materials when there were apparently incorrect practices in new PERS use.

Assistant nurses stressed that existing training focused merely on the general use of PERS, while little was known about how to apply the new PERS in local contexts. In the training session, the hands-on practical exercises were about how to use the system from a technical perspective. For instance, assistant nurses were trained about how to sign in the system, how to answer or confirm one alarm, and which function would be triggered by clicking which buttons. They were trained in a simple and isolated context where assistant nurses deal with one alarm without being disturbed by other things. However, the use of the new PERS in reality included many other contexts. For example, assistant nurses may receive many alarms at the same time or need to simultaneously do tasks for two patients. This led to the result that many assistant nurses lacked competence in making sense of and making the best use of the new PERS in their daily work. In nursing home 1, assistant nurses talked about their feelings with the training session: “*It (the training session) was mainly about how the PERS generally works. But what are the meanings of each step in my work? How do the functions work in our environment? [01A03]*” and “*there was no information (in the training session) about, such as the system volume can be adjusted to prevent disturbing others [01B04]*”, and thereby their abilities to make the best use of the PERS were limited. However, less confusion of technology application in daily work was found in nursing home 2. Their middle managers explained that they did find similar problems at the beginning, but then they actively followed their assistant nurses for one week to address these problems. They pointed out that staff training should not just rely on service providers since knowledge of system application was also based on the understanding of local environments: “*We took for granted that the service provider is professional in training people in how to use the system. But the applied use of the system needs information about our local environments [02B01]*”.

### 4.2. System

#### 4.2.1. Limited System Readiness

Unanticipated system problems emerged during the process, despite the expectation that there should be no technical problems in using the PERS (Table 4). In these two nursing homes, over one-fifth of assistant nurses reported that they met technical problems. Additionally, almost half of them disagreed that the system was compatible. They described besides technical issues such as signal disconnections, problems also included incompatibilities between the PERS and the environment. As we observed, middle managers called service providers frequently to solve problems with the PERS. Some problems required several days to solve. Assistant nurses illustrated that the new system presented conflicts with their work: “*Question: As you said, the system does not adapt to your work. Can you explain more? Interviewee 01A04: For example, I have to deal with many cases at the same time, but this system assumes I do cases one by one. So it is impossible to answer to alarms at the same time.*”

#### 4.2.2. Reasons for Limited System Readiness

Regarding the question about why the system was not ready, the reasons were the same in both nursing homes. This included the lack of ‘mutual communication’ with service providers, and bureaucratic procurement systems (Table 4).

Service providers would solve problems or propose explicit solutions only when the nursing home met very specific problems. Middle managers argued this passive way of communication led to difficulties in getting a full picture of potential problems with PERS. For example, one middle manager said they were not well prepared because of the lack of information about potential technical problems from the service provider: “*I took for granted that the system should have no technical problem when I installed it. But we met, again and again, signal disconnections. These are the things supposed to be solved before implementation, but the company never mentioned them [01B01]*”. Similarly, another middle manager described the problem-solving situation as ‘testing the luck’: “*to be honest, I use the system for solving existing problems and preventing potential problems, not for adding new problems. I am not saying having new problems is not acceptable, but at least, we should be informed beforehand to get something prepared. Not like, we wait and see if we are lucky enough to avoid some problems [02B01]*”.

As there were many unanticipated problems in new PERS, some extra resources such as devices were required. Both assistant nurses and middle managers described the difficulties in getting approvals of sourcing components or technical adaptations to the system, and thought the difficulties led to a disrupted implementation rhythm and limited system adjustments. Middle managers stressed the great uncertainties in the long layer-by-layer approval process: “*For example, we want to connect the PERS with our fire alarms. The required approvals include system procurement, system installation, engineer contract,* etc. *Anything could happen in such a long process [01B01]*”. In this case, middle managers preferred to solve problems or manage technological adaptations depending on the ongoing implementation situation, even if they admitted there was a downside: “*You never know what might happen. It is wiser to wait and see. Though it might need more time and effort to deal with the emergent demands of resources [01B01]*”.

### 4.3. Work Routine

#### 4.3.1. Limited Routine Fluency

The work routines in which the new PERS were embedded were not fluent or unified, despite the expectations that assistant nurses could substitute the old system gradually until the new PERS became a core part of daily work (Table 4). The limited routine fluency was observed in both nursing homes. According to our survey, the rates regarding the perceived system integration decreased slightly. Some assistant nurses wrote reflections in the questionnaire that “*having the older system and the new system at the same time causes much confusion*”. This was consistent with the observation notes. Some assistant nurses strongly relied on the old system and missed alarms from the new PERS. The practice of using the new PERS as a core part of daily work also differed individually.

#### 4.3.2. Reasons for Limited Routine Fluency

Assistant nurses from both nursing homes attributed the limited routine fluency to the ambiguities in individual responsibilities and collective practices of using the new PERS (Table 4). However, participants from nursing home 1 emphasized the ambiguities in individual responsibilities while those from nursing home 2 emphasized the ambiguities in collective practices. It was acknowledged in both nursing homes that assistant nurses had heavy individual responsibilities to ensure the new PERS worked normally: “*we can decide what to do with the system. We are told to keep doing the things that we are doing or what we think is right [01A12]*”. However, great responsibilities could also be problematic. First and foremost, there existed confusion in “*what assistant nurses should and should not do, in terms of integrating the new system in their work [01B02]*”. For example, having the new PERS in parallel with an old system led to numerous duplicated alarms. This means for the same case, assistant nurses may receive one alarm from the old system and one alarm from the new system. As assistant nurses were responsible for answering all incoming alarms, they had to pay extra attention to problems such as which alarm was duplicated, and which alarm was not. However, the duplicated alarms would continuously come if older adults kept using the old system and the new system at the same time. To solve the problem, there should be a person taking the responsibility in educating older adults to use the new system and encouraging them to abandon the old system. In nursing home 2, middle managers were assigned to take this responsibility. But assistant nurses from nursing home 1 were unsure whether they should take on that role because clear information was lacking about who is responsible for resident education. Besides, heavy individual responsibilities also led to a lack of confidence and a lack of unified personal performance. Assistant nurses admitted their practice of using the new PERS was not always correct: “*don’t know if our practices are correct or not. So far I have to judge and change my practices myself [01A05]*”. The lack of confidence in the new PERS led to the assistant nurses using the new system in different ways: “*I am more confident in the old system because everything related is clear. I also think everything related to the new system is clear. But my colleague doesn’t think so. So we use the system differently. [01A07]*”.

Assistant nurses from these two nursing homes showed their concerns about the shared understanding of how the new PERS, as a core part of daily work, should be sustainably used. Participants from nursing home 1 appreciated the local routine guideline and thought it helped them to be consistent in their collective practices. However, this type of guideline was lacked in nursing home 2, where many uncertainties were observed in using the new PERS on a collective level. Middle managers had realized that allowing assistant nurses to use PERS according to personal interpretation led to potential chaos and risks in a group. For example, the alarm phone should be held at all time by at least one assistant nurses. But people behaved differently in “*when and how to give the alarm phones to their colleagues [02B03]*”. Some assistant nurses preferred to put the phone on the kitchen table or charging stations, while some preferred to give the phone to their colleagues face-to-face. Given there was no consistent way of giving alarm phones to colleagues during the changing of shifts, many assistant nurses could not get the phone in time. This led to the risks that alarms were missed during the shift exchanging.

### 4.4. Duration

#### 4.4.1. Longer Duration of Implementation

Both nursing homes extended their planned implementation duration. The plan was to leave two weeks for the implementation project, but the entire implementation process lasted for three months (Table 4). During the implementation process, we observed that middle managers poured great effort into shortening the time, but inevitably the duration had to be extended.

#### 4.4.2. Reasons for a Longer Duration of Implementation

It was acknowledged in the two nursing homes that the extended implementation time was primarily due to inefficient information distribution and little space for plan revision (Table 4).

Some assistant nurses reported that they did not receive training notice or the information regarding responsible persons in time. Middle managers primarily delivered the implementation information through emails and verbally: “*sending emails is the most efficient way to reach all staff as many assistant nurses have totally different working periods [01B03]*”. However, some assistant nurses, especially those working during the night or who don’t check their emails, relied greatly on face-to-face communication with colleagues. Hence, this group of people was easily left behind if their colleagues did not mention training information. This led to more time being required to distribute the key information among assistant nurses.

Another issue was that the initial implementation plan left little space for potential revision. Middle managers explained that during the implementation process, many outcomes were not achieved as expected. Through collecting feedback from assistant nurses, they realized some problems came from incorrect plans at the beginning. According to our observation, for instance, one middle manager spent three days in communicating with trainers about training details, booking training rooms, confirming participants, printing training materials, and writing down training reports. The work, however, was only briefly mentioned in the implementation plan in one sentence: “get all staff well trained”. Most managers mentioned this point in their interviews: “*very often that in your to-do list, it is only one or two bullet points, and my leader and also myself think it is quick to finish the task. But in the end, you will find you are too enthusiastic with the tasks. To achieve the goal, many things need to be done and you have to change initial plans sometimes [01B02].”* Therefore, some revisions in the initial implementation plan were required. However, as the need for potential revision was not included in the implementation project planning, more time had to be spent making decisions about dealing with emerging problems: “*when estimating the implementation time, we didn’t see so many work and potential problems, so we decided two weeks might be enough. But now we have to pay more effort than expected to achieve the initial project expectations. And that is why we keep extending the time [01B03]*”.

## 5. Discussion

In this study, we witnessed both positive and negative outcomes in the implementations in two nursing homes. We argue, although several expectations were achieved as planned, the implementation would have been better if the identified discrepancies in implementation did not occur or were eliminated in time. Hence, this section discussed how to decrease the identified discrepancies as much as possible.

According to the findings, the main domains where discrepancies occur included staff competence, system adjustment, work routine, and implementation duration. The identified discrepancies mainly attributed to management issues such as poorly-designed training sessions, bureaucratic procurement systems, ambiguous individual and collective responsibilities as well as the lack of consideration about invisible work. To narrow the discrepancies between expected and actual implementation, we suggest it is important to prepare for implementation, adjust to the new practices of using the technology, and leave space and time for facilitating the implementation.

### 5.1. Preparing for Implementation

To ensure technology integration has a good start, our findings highlighted the importance of taking account of the interplay between technology and the current social environments, perhaps through fully communicating with service providers, pre-testing the new system, and involving assistant nurses in planning implementation projects. The social environments are defined as the environments encompassing ‘the immediate physical surroundings, social relationships, and cultural milieus within which groups of people function and interact’ [57]. This referred to, for instance, the division of labor in the implementation process and the organizational regulations about work routines in our investigation. Drawing on insights from domestication theory, we argue that a relatively fluent technology integration process requires pre-considering the interplay between technology and current social environments. In this study, issues in preparatory work were represented as insufficient training content, ambiguous individual and collective responsibilities, and unrealistic estimates of implementation duration.

The scripts integrated in the technology show how it is supposed to be used [41]. Hence, the introduction of PERS brought a new way of thinking and acting, affected existing social relationships, and established norms such as individual responsibilities. In turn, the social environment also shaped the functions and scripts of PERS. This finding has been supported by many recent publications [12,27,58,59]. During the ‘shaping and being shaped’ process, PERS required changes in its functions and scripts, while the existing social environments also required adjustments. In our study, this is why some system problems emerged, and why assistant nurses felt uncertain about the system values and their responsibilities in daily work. Additionally, as described, the implementation outcomes from the two nursing homes were mostly the same, but there was one slight difference in terms of the work routine fluency. A possible reason for that was the different division of labor within the social environments. Referring to our collected data, the middle managers from one nursing home had a clear mind that they were responsible for resident education, while those ones from another nursing home were not sure whether they should take this responsibility or not. We argue that taking account of the interplay between technology and current social environments can help to ensure sufficient preparation, and thus quickly meet various needs emerging during implementation, thereby promoting the integration process. Hence, some requirements for adjusting technology and social environments could be quickly fulfilled through the preparatory work in implementation plans.

### 5.2. Adjust to New Practices

To ensure technology integration moves in the right direction, it is important to adjust to the new practices of using technology in daily work, perhaps through establishing clear organizational monitoring directives or having regular auditing meetings. Our findings have illustrated that more demands emerged than were planned for because of the dynamic technology integration process. However, these demands were not met due to the lack of corresponding directives. In this study, the fact that assistant nurses were uncertain about their practice of using the new PERS and there was a lack of directives for adjusting to the new practice proved to be obstacles to the uptake of PERS in daily routines.

As described in Section 5.1, sufficient preparation before implementation is important. However, it should be understood that the negotiation between technology and existing social environments is dynamic and non-linear [18,60], therefore it is difficult to predict all problems [61,62]. Assistant nurses and middle managers highlighted unanticipated system problems, ambiguous system values, and unclear responsibilities regarding using the new PERS. For instance, as reported in our study, there were various personal interpretations in making sense of PERS and confusion in ‘when to do what’ in work routines. Therefore, instead of merely focusing on solving specific problems, there needs to be a way of adjusting to new practices after the introduction of technology. In this study, while middle managers sought direction and resources from the service provider, assistant nurses were already using the new PERS in unexpected ways. They tended to digest most problems and configured the practices regarding PERS use through personal interpretation. The finding of reconfigurations between users and technology is consistent with many studies [63,64,65]. In one of our observed nursing homes, less incorrect use of technology was observed because the middle managers followed their assistant nurses for one week, with a focus on correcting the practices of PERS use. These results point at the importance of tracking users’ practices for addressing personal incorrect ways of using technology. Additionally, despite the fact that learning from colleagues is common among assistant nurses, this might be problematic because there is a risk of learning incorrect knowledge. Our findings support this argument, we further note that assistant nurses indeed recognize the shortcoming of learning from colleagues, so they tend to use documents such as training materials for compensation. Having a monitoring and auditing mechanism to adjust to new practices might help implementers to identify unanticipated problems and solve them in time, and help users to be clear about how to apply new technology in their work. Therefore, adjusting to new practices will help to eliminate uncertainties in the implementation process, and also ensure the coherence between technology and current social environments.

### 5.3. Leaving Space and Time for Facilitating Implementation

To ensure successful technology integration, the considerations on facilitating measures matter. According to Latour [66], the integration of new entities is a question that a ‘collective’ is to be ‘assembled’ to compose a ‘common world’. Namely, even if the technology and the social environments are perfectly adjusted, work is required to make them integrated [44,67]. Our data as a whole have illustrated the work is easily overlooked when making implementation plans. In this study, the invisible work referred to the tasks that deal with unexpected situations such as revising non-influential training materials, contacting service providers for technical problems, waiting for approvals of resource procurement, and distributing key information to all staff.

Middle managers highlighted there was a grey area between expectations and planned inputs or activities, making it difficult to operationalize implementation plans in practice efficiently. The result is consistent with existing studies [68]. For instance, simply adding expectations such as “getting all staff well educated” may overlook numerous invisible work, such as sending out training notices, booking training rooms, printing training materials, ensuring staff gained enough knowledge, and looking for useful learning approaches for future staff. In our study, although middle managers tried to build up the links with all assistant nurses, the unsuitable ways of information delivery led some assistant nurses not being informed. To ensure everyone received the key implementation information, middle managers had to do extra work to educate assistant nurses. Overlooking these invisible facilitating measures made it difficult for implementers to clarify their division of labor. The lack of efficient information delivery approach has been well discussed as a significant barrier to technology uptake and integration. Scholars suggest that introducing new technology should consider how individuals currently acquire information and the ways that they treat different types of materials [69,70]. Similar studies also emphasize the importance of clear leadership in technology implementation [18,71]. Summarizing, there is much invisible work that could not be predicted completely before implementation. To deal with this, leaving space and time matters. The adequate space and time could decrease the urgency of finishing invisible work among middle managers, and thus help to achieve the initial project expectations smoothly.

### 5.4. Limitations

In this study, some limitations existed. First, given our small number of participants, there may be other discrepancies that were not observed. Additionally, the diversity of participants should be considered as it may affect the findings. As described, assistant nurses acknowledged that residents used the PERS efficiently and quickly. The reasons could be that many older adults have good technology experience, or that they received adequate resource support during the implementation. But because we lacked contact with the residents, it was difficult to make conclusions. Unanswered questions also included whether there are differences in assistant nurses’ reflections after long-term use of the new PERS.

## 6. Conclusions

This study identified and analyzed the discrepancies between expected and actual implementation of PERS in two nursing homes. Our findings demonstrate that the complexity of the technology integration process was underestimated. The uptake of technology in daily practice is far more nuanced than a technology implementation plan might imply. The reasons for these discrepancies were multiple. It is important to prepare for implementation, adjust to new practices, and leave space and time for facilitating implementation. Future studies are needed to cover a broader range of participants or track longer-term use of technology to gain a deeper understanding of the entire map of technology implementation.

## Figures and Tables

**Table 1 ijerph-17-04245-t001:** Characteristics of selected nursing homes.

Items	Nursing Home 1	Nursing Home 2
**Characteristics (number)**		
Units	6	6
Seats	59	54
**Unit Functions (number)**		
Units for Dementia	2	2
Units for Disabilities	4	4
**Organizational Structures (number)**		
Assistant Nurses	42	37
Middle Manager	3	2
Residents per Unit	9–10	9
Assistant Nurses per Unit per Shift	2–3	2–3

**Table 2 ijerph-17-04245-t002:** Summary of collected data.

Time	Data Collection and Participants	Aims
August 2019 to September 2019	1 in-depth interview with 1 nursing home leader (40 min)	To draw a logic model about implementation expectations
	Documents analysis of implementation materials	
September 2019	Pre-implementation questionnaire (57/79 assistant nurses)	To ascertain the actual situations
January 2020	Follow-up questionnaire (46/77 assistant nurses)	
October 2019 to December 2019	6 interviews with 4 middle managers (21–45 min each) 7 interviews with 7 assistant nurses (21–45 min each)	To explain the occurrence of discrepancies
	Observation of middle managers and assistant nurses (67 h)	

**Table 3 ijerph-17-04245-t003:** Planned inputs and activities and expected outcomes according to the logic model.

Planned Inputs	Planned Activities	Expected Intermediate Outcomes	Expected Long Term Outcomes
Start-up funding One training session from the service provider Training materials such as instruction manuals Default alarm devices such as alarm phones and buttons IT support group from the service provider Two weeks for the implementation	Send emails to ask all assistant nurses to join the training session Keep the IT support group in contact Replace the old system by the new PERS gradually Extend the management of system implementation to middle managers temporarily	Staff Competence	Improved patient safety Improved user satisfaction High perceived system usability
		All assistant nurses gain adequate knowledge through training	
		Useful training materials for consolidation of knowledge	
		System	
		A non-technical-problem system	
		A compatible system	
		Work Routines	
		A fluent routine embedded with new PERS	

**Table 4 ijerph-17-04245-t004:** Identified discrepancies between expected and actual implementation, and corresponding reasons.

Expectations	Discrepancies	Reasons
Staff Competence	Limited Staff Competence	
Adequately trained assistant nurses	23% of assistant nurses need more knowledge in system use	The complexity of training materials
Useful training materials	5% of assistant nurses used instruction manuals	Lack of training on how to apply the system in local contexts
System	Limited system readiness	
A non-technical-problem system	29% of assistant nurses met technical problems	Lack of ‘mutual communication’ with service providers
A compatible system	49% of assistant nurses disagreed the system was compatible	Bureaucratic procurement systems
Work routines	Limited routine fluency	
A fluent routine embedded with new PERS	Assistant nurses rated ‘system being a part of my work’ decreased from 3.29 to 3.05	Ambiguities in individual responsibilities
	Assistant nurses confused with two systems	Ambiguities in PERS collective use
Duration	Longer duration	
Two weeks	The implementation lasted 3 months	Inefficient information delivery
		Little space for plan revision

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
