# Peer review of "Discrepancies between Expected and Actual Implementation: The Process Evaluation of PERS Integration in Nursing Homes"

_ijerph, 2020, doi:10.3390/ijerph17124245_

Round 1

Reviewer 1 Report

Thank you for the opportunity to review this article.

It looked to be a very promising contribution to the literature, with a strong justification for its approach and good theoretical underpinning, well supported by an impressive literature. The methods used were also impressive, involving both qualitative and quantitative approaches and promising to produce interesting results. Unfortunately, as I read the article, I was disappointed that there was no comparison between the two care homes, which could have really made its arguments about its findings more convincing. As it stood, I had doubts about the presentation of its results and conclusions. These are expressed in the following notes:

Abstract – This needs to be revised in accordance with the comments below

Introduction

Well written

Line 50 – goal ‘was’ that

Line 66 – How – but also why? How tends to be descriptive, while why is more explanatory

Theoretical framework

Well-written

Line 95-96 ‘an integrated component in a system of care practices?’

Line 97 Silverstone and Hirsch

Line 107 – typo

Methods

It was disappointing not to have any reference to a specific methodology – would you say you were using a case study methodology (R.K. Yin 2009) for example, or mixed methods research (MMR)? The justification for your choice of method should also be presented.

Line 113 – ‘out of’ = ‘’from’

Line 117 – ‘met’… ‘to obtain’

Table 1 – very poorly set out – The columns should be headed ‘number’, and ‘size’ should be replaced with ‘characteristics’

Data collection

When did each intervention take place and how long did the intervention take?

Line 130 – ‘were confirmed’ – what does this mean – were they formal collaborators in the research, or partners in kind, did they give formal consent? As for the assistant nurses – did they go through a recruitment process and if so what was it? What is the definition of ‘assistant nurses’ in your context? Are there clinically trained ‘nurses’, senior nurses’, and then ‘care assistants’?

Line 132 – I’m not sure about the use of the phrase ‘stopping criterion’ in this context – isn’t it used in mathematical algorithms rather than qualitative research?

Table 2 – might be a good place to indicate the intervention start and duration. The table is confusing – there is only 1 nursing home mentioned here? The interviews should be grouped together in one section, and the questionnaire returns (ending Jan 2020) should be the last item to be tabulated.

Line 139 – topic guide – rather than ‘guideline’

Line 140 – ‘was proposed’ – inappropriate use of language; ‘The interview started with an open question….’

Line 146 - interview notes rather than ‘content’ – and how could you transcribe it if it wasn’t recorded?

Line 149 – In this part of the study

Line 172 – the observations reported here appear to be akin to an ethnographic approach. If this is not so, is there a reference you could use to support your choice of method?

Line 185 – was the verbal consent recorded? Were they provided with a participant information sheet? Did this study undergo ethical review?

Line 188 - inappropriate use of language ‘contained’ – ‘consisted of’. A logic model as I understand it is usually diagrammatic not tabular. Where is it? Table 3 refers to the comparison and reference to it should come in line 191.

Line 191 – Lastly

Table 3

‘Extend the mandate of system implementation to middle managers temporarily’ – what is meant by this? Do you mean ‘management’ rather than ‘mandate’?

‘Expected intermediate outcomes’ referred to expected direct outcomes of the inputs and activities;

So why are these not called ‘expected direct outcomes’ to reduce confusion?

“‘Expected outcomes’ referred to expecting benefits or disadvantages resulting from intermediate outcomes”

This could just be called expected benefits or disadvantages?

In this way, you would not have to explain it because the terms are clear enough to be understood by the reader.

Line 204-206 – do you mean that you are focusing on reporting on the ‘discrepancies’ in this paper – otherwise why collect data on the benefits and disadvantages?

3.3.3 – there needs to be some clarity in your use of the terms ‘concepts’, ‘codes’ and ‘themes’.

3.4 Ethical considerations should ideally be made clear earlier on in the methodology. Ethical issues that arise in the course of the research should be reported in the Findings/Results.

Participants don’t usually get ‘coded’, unless each one was actually given a code to themselves? – do you just mean they were given a participant identifier? Because their transcripts are usually just stored in the documents section of NVIVO, and not coded.

Table 4 - inappropriate use of language:

Lack of ‘mutual communication’ with service providers - Difficulties in resource allocation

Line 251 – needed

Line 254 – did they deliberately keep the alarms muted or was there some technical misunderstanding involved? Did they change their behavior on realizing this was happening?

Line 257 – This included….

Line 259 - Assistant nurses tended to learn from colleagues – was this observed? This line should be linked with line 265 where you explain this….

Line 268 – ‘in the same way’ as who?

Line 269-271 – you might want to distinguish between knowledge, and practice, or the practical application of knowledge; it’s interesting that she felt every detail of information wasn’t required – for the technology to be practically useful. What are the implications of this attitude?

Line 272 – do you mean ‘general’ rather than ‘literal’? Were they not given some time to trial/test/pilot the technology – were they not given any hands-on practical exercises to help them handle the equipment in their training? Was the training observed in the research? I suggest this was the problem rather than understanding ‘local environments’ – what is meant by local environments? In any other environment, the inability to adjust the volume is still a hindrance. Line 282-292 is more appropriate in illustrating this point about local environments.

‘Besides technical issues such as signal disconnections, problems also included incompatibilities between the PERS and the environment’ – this sentence belongs in 4.2.1

4.2.2 The title is about staff competence, but the paragraph below speaks to me of failings on the part of the service provider and the technology

Para 309-318 I’m not sure this describes ‘resource allocation’ – the quotes here seem to me to be about independent problem-solving or managing technological adaptations to the system which includes getting approvals for these adaptations, sourcing components etc – and have little to do with staff competence, but rather with bureaucratic systems of governance? Or is this just about the problems of sourcing component parts to deal with technological problems? These are limits to staff ability to problem-solve.

Line 330 and 332 – ‘personal responsibility’ is not a term I would use – ‘a heavy individual responsibility’ is preferable. Were there conflicts between individual and collective roles and responsibilities for the care of residents?

Line 339 – were the alarm sounds not distinguishable?

Line 341 – ‘taking the’

Line 342 – it’s about being not having been assigned a role – this is a management issue

Line 347 – ‘using’ the new system; The quote which follows seems to tell me that the assistant nurse is basically not convinced of the value of the system. There also hasn’t been any discussion of the time needed to adapt to the system nor to explain the system to the residents

Line 351 – ‘Many’ rather than ‘Great’

Line 352 – ‘had’ realized

Line 354 – 7/24 – do you mean 24 hours 7 days a week?

Lines 353-356 – unclear in meaning

Lines 366ff – shortened forms such as ‘didn’t’ are not really acceptable

Line 375 – ‘illustrated’ – explained, you mean

Lines 385-389 – this is new information which should have been in the results section. The discussion section should be discussing the reported findings

Line 389 - ‘thought’ the new PERS

‘In our view, the implementation might be improved if the identified discrepancies in implementation did not occur or were eliminated in time.’ – that’s stating the obvious!

Line 404 – repetitive

Line 408 – how were existing social relationships affected? Not evident enough from your results

Line 409 – what is this ‘social environment’ you refer to? There was only data from one care home – if there were two care homes represented, comparisons showing the differences between them would have explained the importance of the ‘social environment’ better, if the experience of the application of the technology was different in different environments.

Line 422 – monitoring activities, not audit. And the problem is in poor management, not just directives

Line 437-438 – this to me appears to point to individual agency, using individual initiatives which should be affirmed - or not?

Line 451 – ‘was relevant’ – not clear what is meant

Lines 457-459 – were these observed? If so, why were they not reported and discussed in the results?

Lines 468-470 – not clear in meaning. Do you simply mean that invisible work will be reduced if there was enough preparation and training before implementation?

Lines 476-477 – this interesting bit of new information should be in your findings and discussed there.

Appendix

Poorly devised Table - the first few rows do not make sense. ‘number of years’; explanation of scores is needed

Reviewer 2 Report

Introduction can be improved with more results of PERS applications to compare with this study, considering the small number of participants involved.

Could be more described the qualitative method of “content analysis” used ( NVIVO 12).

Author Response

Please see the attachement. Thank you!

Reviewer 3 Report

Title: Discrepancies between expected and actual implementation: the process evaluation of PERS integration in nursing homes

This is a well written manuscript. Some minor suggestions for improvement:

  • Table 2. is a little bit confusing for the reader. Suggest to restructure the table content to include here the response rates (total or per nursing home)
  • Lines 137-147 – why only 1 leader (from 1 nursing home) was interviewed regarding the expectation? Shouldn’t it be listed as an additional limitation?
  • Lines 158-163 – suggest to provide direct references to the questionnaires used/combined also here in the text (not only in the Appendix). Does ‘well established’ equals validated? As there are some additional (added by the research team) questions – was the questionnaire pre-tested?

Round 2

Reviewer 1 Report

Thank you for working hard to improve the manuscript.

Please find attached my comments and changes in red in your responses to my initial review comments. I hope you will find them useful in further improving the paper. I like the fact that your paper highlights the 'invisible work' that staff have to undertake in order for a technological intervention to work successfully, and also how the relational work of getting help and coordinating work with colleagues is so important. 

All the best for your publication.

Author Response

Dear reviewer,

Thank you very much for your patience and valuable suggestions. We have revised the manuscript carefully. Please see the attachement. 
